# Knowledge-Augmented Language Model Verification

**Jinheon Baek   Soyeong Jeong   Minki Kang   Jong C. Park   Sung Ju Hwang**
KAIST
{jinheon.baek, starsuzi, zzxc1133, jongpark, sjhwang82}@kaist.ac.kr

## Abstract

Recent Language Models (LMs) have shown impressive capabilities in generating texts with the knowledge internalized in parameters. Yet, LMs often generate the factually incorrect responses to the given queries, since their knowledge may be inaccurate, incomplete, and outdated. To address this problem, previous works propose to augment LMs with the knowledge retrieved from an external knowledge source. However, such approaches often show suboptimal text generation performance due to two reasons: 1) the model may fail to retrieve the knowledge relevant to the given query, or 2) the model may not faithfully reflect the retrieved knowledge in the generated text. To overcome these, we propose to verify the output and the knowledge of the knowledge-augmented LMs with a separate verifier, which is a small LM that is trained to detect those two types of errors through instruction-finetuning. Then, when the verifier recognizes an error, we can rectify it by either retrieving new knowledge or generating new text. Further, we use an ensemble of the outputs from different instructions with a single verifier to enhance the reliability of the verification processes. We validate the effectiveness of the proposed verification steps on multiple question answering benchmarks, whose results show that the proposed verifier effectively identifies retrieval and generation errors, allowing LMs to provide more factually correct outputs. Our code is available at https://github.com/JinheonBaek/KALMV.

## 1  Introduction

Recent Language Models (LMs) (Brown et al., 2020; Chowdhery et al., 2022; Chung et al., 2022), which have a large number of parameters and are further instruction-finetuned on massive datasets, have achieved remarkable successes on various language tasks. For example, they are able to perform closed-book zero-shot question answering, which aims to provide an answer to a user's query without

updating the LM parameters while using only the knowledge internalized in their parameters. However, while the generated answers from LMs look plausible and sound, they are often factually incorrect, which is a problem widely known as *hallucination* (Rohrbach et al., 2018; Bang et al., 2023; Zheng et al., 2023). Hallucination is a critical problem when deploying LMs, since it poses a risk of spreading misinformation, potentially misleading users who rely on the information.

To mitigate hallucination of LMs, recent works have proposed to augment LMs with the knowledge retrieved from external knowledge sources (e.g., Wikipedia and Wikidata) (Lazaridou et al., 2022; Mallen et al., 2023; Baek et al., 2023). Moreover, some other works have proposed to check the factuality of generated texts and refine them by using the knowledge in LMs themselves or from the external knowledge sources (Madaan et al., 2023; Gao et al., 2023; Jiang et al., 2023; Gou et al., 2023; Xu et al., 2023; Feng et al., 2023). However, while the aforementioned knowledge-augmentation strategies are effective in reducing hallucinations, we find that there still exists a couple of challenges: 1) the retrieved knowledge may not be relevant to the given question from the user, and 2) the generated answer may not be grounded in the retrieved knowledge, as illustrated in Figure 1 and shown in Figure 2.

In this work, we aim to overcome these suboptimalities of knowledge-augmented LMs. In other words, our goal is to verify whether the retrieved knowledge used for augmenting LMs is related to generating the answers for the given questions and whether the generated answers include the relevant parts of the retrieved knowledge. To this end, we propose to train a small, tailorable LM that is able to verify the aforementioned two failure cases of knowledge-augmented LMs in retrieval and generation steps. More specifically, we first automatically construct the training labels by categorizing the failure of knowledge-augmented LMs into two

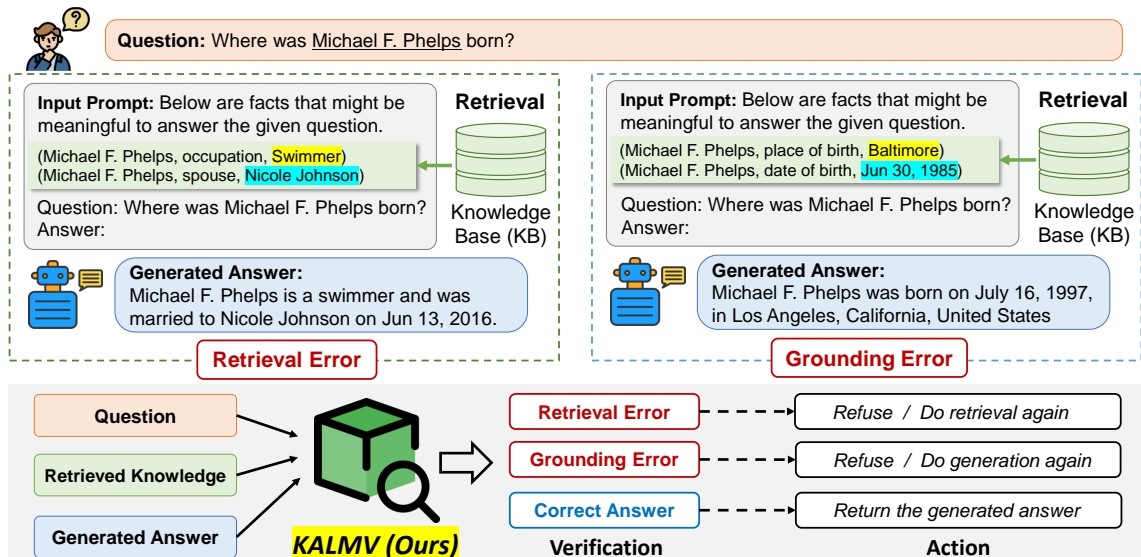

Figure 1: Existing knowledge-augmented language models first retrieve the relevant knowledge to the given query from the external knowledge base and then augment the LMs with the retrieved knowledge to generate the factually correct responses. However, there are two types of common errors: 1) the retrieved knowledge might be irrelevant to the given query (retrieval error); 2) the generated answer might not be grounded in the retrieved knowledge (grounding error). Our proposed KALMV can detect those two types of errors in knowledge retrieval and grounding, and also iteratively rectify them, reducing hallucinations.

cases: retrieval error and generation error, based on the triplet of the input question, retrieved knowledge, and generated answer. Then, we instruction-finetune the LM with pairs of a certain verification instruction and its associated label, during verifier training. At the inference step, we validate the generated texts through our verifier, to filter out potentially incorrect generations due to retrieval or generation failures, to prevent the generation of texts with inaccurate information. Note that there exists a concurrent work (Peng et al., 2023) that proposes to check whether the generated answers from LMs are grounded in the knowledge provided to LMs, by using API calls to proprietary LLMs or a heuristic measure (F1). However, this work clearly differs from our method, since we further verify the **relevance of the retrieved knowledge** in addition to the answer groundedness, through instruction-finetuning of LMs.

In addition, we further propose refining the output from knowledge-augmented LMs if our verifier identifies the error in either the knowledge retrieval or the knowledge reflection. Specifically, we repeat the answer generation process until the model retrieves the knowledge relevant to the given question and incorporates the correctly retrieved knowledge into the generated answer, based on the verifier outcome. Also, since detecting errors of knowledge-augmented LMs with a single instruction given to the verifier might be inaccurate, we further construct **an ensemble over multiple outputs from different instructions** with a sin-

gle verifier. Notably, one extra advantage of our verifier is that it is a plug-and-play module that works with any public or proprietary LMs, since we only require input-output pairs of LMs for verification without any architectural changes. We refer to our proposed method as **K**nowledge-**A**ugmented **L**anguage **M**odel **V**erification (**KALMV**).

We experimentally validate the effectiveness of our KALMV on two different Question Answering (QA) tasks, namely open-domain QA and knowledge graph QA. The experimental results show that our KALMV can effectively verify the failure cases of knowledge-augmented LMs in knowledge retrieval and answer generation steps, contributing to significant reduction of the hallucination. Also, further analyses demonstrate the effectiveness of our error-rectifying and ensemble strategies.

Our findings and contributions are threefolds:

- We point out the underexplored challenges of knowledge-augmented LMs, which are retrieval of irrelevant knowledge and unfaithful knowledge grounding.
- We introduce a novel verifier that identifies whether the retrieved knowledge is relevant to the question and reflected in the answer, and further present useful strategies for rectifying incorrect answers as well as improving the effectiveness of the verifier via ensembling.
- We validate our KALMV on open-domain and knowledge graph question answering tasks, demonstrating its effectiveness in verifying the errors of knowledge-augmented LMs.

## 2 Background and Related Work

**Language Models** Pre-trained Language Models (LMs) (Devlin et al., 2019; Liu et al., 2019; Radford et al., 2018; Raffel et al., 2020), which are trained on a large corpus with self-supervised learning, show impressive performances across diverse natural language tasks and are used as the base architecture. Recently, large language models (Brown et al., 2020; Chowdhery et al., 2022; Touvron et al., 2023) having billions of parameters are able to respond to a user's query without any model training on the target task. On the other hand, finetuning LMs on a massive collection of natural language datasets phrased as instructions (Wei et al., 2022; Chung et al., 2022; Sanh et al., 2022), which is known as instruction finetuning, also enables the LMs to attain reasonable zero-shot learning abilities without focused training on the target task. However, while large and instruction-finetuned LMs show performance improvement on factual tasks (e.g., question answering), they are still suboptimal since they cannot memorize all the world knowledge and may contain distorted facts. To overcome this challenge, recent studies propose augmenting LMs with external knowledge, which we discuss below.

**Knowledge-Augmented LMs** Early works aim to incorporate knowledge from external knowledge sources (e.g., Wikipedia) into LMs, in order to enhance their performances on tasks that require factual knowledge, such as question answering. While such previous knowledge-augmented LMs (Zhang et al., 2019; Guu et al., 2020; Yamada et al., 2020; Qin et al., 2021; Borgeaud et al., 2022) show performance improvements on knowledge-intensive tasks, in order to integrate the external knowledge, they utilize the specific pre-training but also require changing the model architecture, which are not easily generalizable across different LMs and tasks. Similarly, while some recent works (Lewis et al., 2020; Kang et al., 2022; Li et al., 2022; Izacard et al., 2022) propose augmenting LMs with external knowledge during finetuning, they also require specific training on each target task and dataset, and often require architecture modifications. However, training the task- and data-specific LMs with model updates are computationally prohibitive as the size of LMs increases exponentially. Also, previous approaches involving architecture changes are not applicable to black-box LMs (e.g., ChatGPT), which are accessible only through API. Considering these challenges, recent methods (Lazaridou et al., 2022;

Trivedi et al., 2022; Baek et al., 2023; Shi et al., 2023; Peng et al., 2023) use the large or instruction-finetuned LMs to incorporate the external knowledge, which allows us to design only the input text to LMs without requiring additional training thanks to their strong generalization capabilities. Following this trend, we focus on knowledge-augmented instruction-finetuned LMs, while exploring their two underrepresented challenges: incorrect knowledge retrieval and unfaithful knowledge reflection.

**Knowledge-Augmented Fact Checking** Similar to the motivation of the aforementioned knowledge-augmented LMs, recent works (Mallen et al., 2023; Gao et al., 2023; Peng et al., 2023; Jiang et al., 2023; Xu et al., 2023) propose to check the factuality of the answers generated by LMs using the external knowledge. Typically, these approaches generate the answer in response to the user's query with LMs, and then identify whether the generated answer aligns with the retrieved knowledge. However, there are significant differences between our work and the existing methods. First of all, they assume that the retrieved knowledge is pertinent, which is yet unrelated and unhelpful sometimes, making the model generate incorrect predictions. In contrast, our proposed verifier can recognize the relevance of the retrieved knowledge before incorporating it into the LMs. Second, previous works suppose that the retrieved knowledge used for fact-checking is accurately reflected in the generated answer; however, LMs often ignore the given knowledge and hallucinate the answer, whereas we can detect and rectify such the grounding error. Lastly, unlike most fact-checking methods that always provide the answer with its refinement, our method can further decline to provide answers unless they are validated as correct. These differences highlight the novel contributions of our verification approach, compared against previous fact-checking methods.

## 3 Method

We now formally describe knowledge-augmented LMs, and present our method, Knowledge Augmented Language Model Verification (KALMV).

### 3.1 Knowledge-Augmented Language Models

We begin with the explanation of language models.

**Language Models** In our problem setup, the goal of Language Models (LMs) is to generate a factually correct answer in response to an input query from a user, which is formally defined as follows:

$\hat{\boldsymbol{y}} = \text{LM}(\boldsymbol{x})$, where $\boldsymbol{x}$ and $\hat{\boldsymbol{y}}$ are the input and output pair, each of which consists of a sequence of tokens, and LM is the language model. We assume that LMs are already trained on massive instruction-finetuning datasets, which are capable of performing diverse tasks (e.g., question answering) (Wei et al., 2022; Chung et al., 2022), and also not further trainable since we sometimes cannot update the parameters of LMs due to their huge sizes or inaccessibility (OpenAI, 2023; Anil et al., 2023).

Note that, while previous works (Petroni et al., 2019; Roberts et al., 2020) show that LMs are capable of memorizing the knowledge seen during training, such naive LMs encounter several challenges when dealing with factual questions. In particular, LMs cannot memorize all the factual knowledge due to their limited number of parameters. Also, some knowledge is changed and updated over time; however, LMs remain static unless they are further trained while training them is also very expensive.

**Knowledge-Augmented LMs** In order to tackle the aforementioned challenges of naive LMs, some works (Lazaridou et al., 2022; Mallen et al., 2023; Baek et al., 2023) propose to augment LMs with the knowledge retrieved from the external knowledge base, called knowledge-augmented LMs. Formally, let $\mathcal{K}$ be the external knowledge base, which could be an encyclopedia (Wikipedia) consisting of millions of documents or a knowledge graph (Wikidata) consisting of billions of facts. Then, we first retrieve the pertinent knowledge $\boldsymbol{k}$ from the knowledge base $\mathcal{K}$ based on its relevance score to the input query $\boldsymbol{x}$, by using the retriever model denoted as follows: $\boldsymbol{k} = \text{Retriever}(\boldsymbol{x}, \mathcal{K})$ where $\boldsymbol{k} \in \mathcal{K}$. After that, the retrieved knowledge $\boldsymbol{k}$ is incorporated into the input of the LM along with the input query, as follows: $\hat{\boldsymbol{y}} = \text{LM}(\boldsymbol{x}, \boldsymbol{k})$. This knowledge augmentation strategy brings impressive performance improvements on factual language tasks by reducing the hallucination issue of LMs.

However, despite the enormous successes of the aforementioned knowledge-augmented LMs, there exist remaining issues that have largely underexplored. First, the knowledge retrieved to augment LMs might be irrelevant to answer the given question, since the retrieval is not always accurate in real-world scenarios. Second, even if the retrieved knowledge is useful, LMs sometimes reflect the irrelevant part of the retrieved knowledge, or might completely ignore the knowledge and generate the answer based on their incorrect knowledge. In par-

ticular, as shown in Figure 2, there are significant occurrences of retrieval and grounding errors.

### 3.2 KALMV: Learning to Verify Knowledge-Augmented Language Models

To overcome the challenges of existing knowledge-augmented LMs, we propose a novel verification method that identifies not only the relevance of the retrieved knowledge to the input question but also the reflection of the knowledge in the generated answer, which we refer to as Knowledge-Augmented Language Model Verification (KALMV).

**Verification of Retrieved Knowledge** Given the triplet of the input query, the retrieved knowledge, and the generated answer $(\boldsymbol{x}, \boldsymbol{k}, \hat{\boldsymbol{y}})$, we aim to verify whether the retrieved knowledge $\boldsymbol{k}$ is relevant to the input query $\boldsymbol{x}$. Since recent LMs (Wei et al., 2022; Chung et al., 2022) can contextualize multiple sentences and understand their underlying relationships, we use such a small and instruction-finetuned LM to identify the relatedness between the input query and the knowledge. To be specific, we prompt the verifier LM to determine the relevance based on the verification instruction $\boldsymbol{i}$ as well as the input, knowledge, and generated answer triplet $(\boldsymbol{x}, \boldsymbol{k}, \hat{\boldsymbol{y}})$, formalized as follows: $o_k = \text{Verifier}_k(\boldsymbol{i}, \boldsymbol{x}, \boldsymbol{k}, \hat{\boldsymbol{y}})$, where $\text{Verifier}_k$ denotes the LM for retrieved knowledge verification, and $o_k$ denotes its output. Note that we formulate the verification task as a multiple-choice question-answering task, i.e., the verifier should produce either "A" for incorrect retrieval or "B" for correct.

**Verification of Generated Answer** Our next objective is to identify whether the generated answer from LM is grounded in the retrieved knowledge. To achieve this, similar to the retrieved knowledge verification process explained in the above paragraph, we use the separate, small-size, instruction-finetuned LM for answer verification. Formally, given the input query, retrieved knowledge, and generated answer triplet $(\boldsymbol{x}, \boldsymbol{k}, \hat{\boldsymbol{y}})$, as well as the instruction $\boldsymbol{i}$ describing the task of generated answer verification, the verifier LM produces the output token, namely "A" or "B" where "A" represents that the retrieved knowledge is not reflected in the generated answer and "B" represents the vice versa, formalized as follows: $o_y = \text{Verifier}_y(\boldsymbol{i}, \boldsymbol{x}, \boldsymbol{k}, \hat{\boldsymbol{y}})$.

Thus far, we propose to detect the errors of knowledge-augmented LMs in knowledge retrieval and answer generation by using distinct LM-based verifiers. However, it is inefficient to perform two

individual verification processes, since both verification formulations are identical. Also, the knowledge retrieval and answer generation processes are sequential, which means that verifying the generated answer is unnecessary if the retrieved knowledge is irrelevant. Therefore, we further combine two verification procedures into one by changing the task instruction accordingly with the single verification LM (Verifier). Specifically, Verifier produces one among the following three options: A. the retrieved knowledge is not helpful to answer the question; B. the generated answer is not grounded in the retrieved knowledge; C. all the other cases.

**Instruction-Finetuning for Verifier** While recent instruction-finetuned LMs might be capable of performing the proposed verification task, it may be more beneficial to tailor the LM to the verification task through additional instruction-finetuning. To perform this, we require the following input-output pairs: $\{(x, k, y), o\}$, where the input consists of the given question, retrieved knowledge, and true answer, and the output is the verification label which we automatically generate. In particular, we first examine whether the retrieved knowledge includes the correct answer, $y \subseteq k$, as annotated in the training data, and then label it as a retrieval error when the knowledge does not include the correct answer. Similarly, if the retrieval is correct yet the generated answer $\hat{y}$ from $\text{LM}(x, k)$ does not have overlapping tokens with the retrieved knowledge $k$, we label it as the generation error. Finally, for all cases where the generated answer is correct, we label it as correct[1]. Then, by using the inputs phrased as instructions and their corresponding labels, we instruction-finetune the proposed Verifier.

**Ensemble Verification** To identify retrieval and generation errors in knowledge-augmented LMs, we forward the instruction along with the query, knowledge, and generated answer to the verifier. However, it might be inaccurate to determine the errors only with a single instruction, since recent LMs are sensitive even to minor changes in the input prompt (Zhao et al., 2021; Lu et al., 2022; Zhou et al., 2022) and also our small-size verifier LM might not fully understand the given input context. Therefore, we design various instructions, forward them to our single verifier, and ensemble the multiple outputs from the verifier with average.

[1]There might be more sophisticated techniques to automatically assign verifier labels, which we leave as future work.

### 3.3 Strategies for Rectifying Errors of Knowledge-Augmented Language Models

Our verification method provides a distinct advantage in contrast to existing knowledge-augmented LMs and knowledge-augmented fact-checking approaches. That is, existing approaches always provide the answers to users even if they are not reliable; however, our method can withhold the answers if errors are detected by the proposed verifier, which can enhance the reliability and trustworthiness of LM-based systems. However, instead of simply refraining from responding to user queries, it is more worthwhile to rectify errors in the knowledge retrieval and answer generation stages. Thus, we further propose simple yet effective strategies, iteratively correcting errors detected by our verifier.

**Rectifying Errors in Knowledge Retrieval** The retrieved knowledge from the external knowledge base might be irrelevant to answer the question due to the retrieval error, which may mislead LMs to generate an incorrect answer. To overcome this issue, we retrieve the new knowledge iteratively until our verifier confirms that the retrieved knowledge is related to answering the question, for a certain number of times (e.g., ten times). Specifically, the knowledge with the highest relevance score to the question is retrieved, while excluding any knowledge that has been used in the previous iterations.

**Rectifying Errors in Answer Generation** Even though the retrieved knowledge is pertinent to the given question, LMs sometimes ignore the knowledge augmented to them and then generate the answer based on their inaccurate knowledge. To tackle this issue, similar to what we previously did on knowledge retrieval, we iteratively generate the answer until the answer is confirmed by the verifier, for the specific number of times. Note that, in order to generate the answer differently across different trials, we leverage the top-k sampling (Fan et al., 2018) that enables stochastic generation processes.

## 4 Experimental Setups
In this section, we describe the datasets, models, evaluation metrics, and implementation details. We provide the additional details in Appendix A.

### 4.1 Tasks and Datasets
We evaluate our Knowledge-Augmented Language Model Verification (KALMV) on factual Open-Domain Question Answering (ODQA) and Knowledge Graph Question Answering (KGQA) tasks.

**Open-Domain Question Answering** The goal of open-domain question answering (ODQA) task is to generate answers in response to factual questions usually with the relevant knowledge retrieved from the external knowledge source. As the knowledge source, we use Wikipedia which is an open encyclopedia consisting of millions of documents. For datasets, we use Natural Questions[2] (Lee et al., 2019) that is modified from Kwiatkowski et al. (2019) for ODQA and HotpotQA[3] (Yang et al., 2018), both of which are designed with Wikipedia.

**Knowledge Graph Question Answering** In addition to ODQA, we evaluate our KALMV method on knowledge graph question answering (KGQA), whose goal is to answer the questions that are answerable by the facts over knowledge graphs. For datasets, we use WebQSP (Yih et al., 2016) that is modified from Berant et al. (2013) to filter out unanswerable questions, and Mintaka (Sen et al., 2022). Further, for the knowledge source, we use Wikidata which includes billions of facts that are represented as the triplet: (subject, relation, object), and we follow the standard preprocessing setup for KGQA (Saffari et al., 2021; Baek et al., 2023).

## 4.2 Baselines and Our Model

We compare our KALMV against relevant baselines that augment LMs with external knowledge and have strategies to reduce hallucinations. Note that models including verification can refrain from providing answers if the verifier identifies errors.

**Naive Language Models** This baseline uses only the LMs without incorporating external knowledge.

**Knowledge-Augmented LMs** This baseline augments LMs with the knowledge retrieved from the external knowledge base (Wikipedia or Wikidata).

**Adaptive Retrieval** This baseline (Mallen et al., 2023) adaptively augments the LMs by retrieving the knowledge only when the external knowledge is necessary. In particular, if the entity that appeared in the question is less frequent, they retrieve the knowledge and provide it to the LMs. This model, namely **Adaptive Retrieval with Entity**, is applicable to questions that have pre-annotated entities (i.e., KGQA); therefore, we also include its variant, namely **Adaptive Retrieval with Confidence**, that augments LMs with retrieval only when the answer generation probability of naive LMs is low.

[2]https://huggingface.co/datasets/nq_open
[3]https://huggingface.co/datasets/hotpot_qa

**LLM-Augmenter** This baseline (Peng et al., 2023) first augments LMs with knowledge retrieval, and then verifies whether the retrieved knowledge is reflected in the generated answer with Knowledge F1 (Shuster et al., 2021) that measures overlapping terms between the knowledge and the answer. Yet, unlike our KALMV, it cannot identify retrieval errors but also uses a heuristic metric for verification. In addition to the aforementioned **LLM-Augmenter w/ Knowledge F1**, we also include the **LLM-Augmenter w/ Confidence** that verifies the answer based on its generation probability.

**KALMV** This is our Knowledge-Augmented Language Model Verification (KALMV) method, which not only verifies both the retrieval and generation errors with the instruction-finetuned tailored verifier, but also iteratively rectifies errors.

## 4.3 Evaluation Metrics

Following the standard evaluation protocol of generative QA (Mallen et al., 2023; Baek et al., 2023), we use F1 which measures the number of overlapping words between the generated answer and the labeled answer with precision/recall, EM which measures whether the generated answer is exactly the same as the labeled answer, and accuracy which measures whether the generated answer includes the labeled answer. For KGQA, following Baek et al. (2023), we further consider a set of alternative names of the labeled answers available in Wikidata.

## 4.4 Implementation Details

We use the same retriever across different models for fair comparisons. In particular, for ODQA, we use BM25 (Robertson et al., 1994) that considers the term-based matching, following Mallen et al. (2023). Also, for KGQA, we use MPNet (Song et al., 2020) that is based on the dense retrieval, following Baek et al. (2023). For the input prompt to LMs for all baselines and our model, we follow the existing works (Mallen et al., 2023; Baek et al., 2023) which use the simple prompt, such as "Context: {Context}. Question: {Question}. Answer: ". Regarding the LMs to generate answers, we use FLAN (Chung et al., 2022) with three different sizes: Base, Large, and XL having 250M, 780M, and 3B parameters, respectively. In our KALMV, we use the FLAN Base as the verification LM, and we instruction-finetune it with the batch size of 8 and the learning rate of 5e-5 with AdamW (Loshchilov and Hutter, 2019) as the optimizer. In addition, we set the maximum number

Table 1: **Results on Natural Questions and HotpotQA for open-domain question answering and WebQSP and Mintaka for knowledge graph question answering**, with FLAN of different sizes as the LM. We emphasize the best results in bold.

| Datasets | Methods | Base (250M) | | | Large (780M) | | | XL (3B) | | |
|---|---|---|---|---|---|---|---|---|---|---|
| | | F1 | EM | Acc | F1 | EM | Acc | F1 | EM | Acc |
| **Natural Questions w/ Wikipedia** | Naive Language Models | 7.53 | 3.24 | 4.57 | 11.09 | 6.29 | 7.81 | 16.89 | 11.16 | 12.94 |
| | Knowledge-Augmented LMs | 18.06 | 12.30 | 15.26 | 18.61 | 13.74 | 16.40 | 19.03 | 14.13 | 16.90 |
| | Adaptive Retrieval w/ Confidence | 16.70 | 11.02 | 14.07 | 18.16 | 13.07 | 15.60 | 20.89 | 15.76 | 18.28 |
| | LLM-Augmenter w/ Knowledge F1 | 19.58 | 13.56 | 16.81 | 28.53 | 21.22 | 25.32 | 31.00 | 23.06 | 27.59 |
| | LLM-Augmenter w/ Confidence | 19.91 | 14.14 | 17.19 | 20.19 | 14.97 | 18.29 | 22.88 | 17.17 | 20.49 |
| | **KALMV (Ours)** | **52.98** | **42.36** | **50.43** | **56.80** | **46.13** | **53.57** | **67.43** | **58.06** | **63.17** |
| **HotpotQA w/ Wikipedia** | Naive Language Models | 14.25 | 9.68 | 10.36 | 16.80 | 11.78 | 12.41 | 21.97 | 15.06 | 16.22 |
| | Knowledge-Augmented LMs | 31.20 | 22.77 | 25.13 | 33.46 | 25.29 | 27.37 | 35.47 | 27.08 | 29.14 |
| | Adaptive Retrieval w/ Confidence | 26.82 | 19.10 | 21.11 | 26.80 | 19.65 | 21.23 | 29.41 | 21.55 | 23.54 |
| | LLM-Augmenter w/ Knowledge F1 | 32.89 | 23.24 | 26.12 | 39.40 | 28.55 | 31.60 | 46.97 | 34.54 | 37.72 |
| | LLM-Augmenter w/ Confidence | 34.75 | 25.67 | 28.20 | 35.78 | 27.29 | 29.38 | 40.57 | 31.35 | 33.71 |
| | **KALMV (Ours)** | **64.06** | **52.31** | **55.84** | **63.74** | **52.39** | **55.98** | **67.21** | **54.99** | **58.07** |
| **WebQSP w/ Wikidata** | Naive Language Models | 32.53 | 21.35 | 25.78 | 40.33 | 30.08 | 32.74 | 46.20 | 36.43 | 40.11 |
| | Knowledge-Augmented LMs | 53.57 | 43.25 | 53.68 | 42.37 | 26.13 | 62.28 | 49.45 | 36.02 | 59.28 |
| | Adaptive Retrieval w/ Entity | 49.13 | 37.79 | 46.32 | 47.81 | 35.68 | 49.32 | 51.99 | 41.54 | 51.16 |
| | Adaptive Retrieval w/ Confidence | 46.76 | 36.49 | 43.66 | 48.32 | 36.56 | 51.98 | 53.17 | 43.32 | 53.89 |
| | LLM-Augmenter w/ Knowledge F1 | 56.42 | 45.95 | 56.26 | 44.41 | 27.79 | 64.56 | 51.95 | 38.12 | 61.96 |
| | LLM-Augmenter w/ Confidence | 56.62 | 47.33 | 56.36 | 44.35 | 28.79 | 64.47 | 50.63 | 36.62 | 60.67 |
| | **KALMV (Ours)** | **74.31** | **63.92** | **77.78** | **54.79** | **45.46** | **82.71** | **67.10** | **50.81** | **83.21** |
| **Mintaka w/ Wikidata** | Naive Language Models | 16.16 | 8.53 | 10.59 | 20.90 | 12.83 | 14.46 | 26.99 | 19.08 | 21.22 |
| | Knowledge-Augmented LMs | 24.28 | 15.46 | 19.15 | 24.57 | 15.39 | 23.77 | 27.74 | 18.23 | 22.92 |
| | Adaptive Retrieval w/ Entity | 23.66 | 14.68 | 17.87 | 25.96 | 16.45 | 22.92 | 30.34 | 21.36 | 24.20 |
| | Adaptive Retrieval w/ Confidence | 21.46 | 13.15 | 16.06 | 25.34 | 16.28 | 22.07 | 29.00 | 20.68 | 23.70 |
| | LLM-Augmenter w/ Knowledge F1 | 27.99 | 18.18 | 22.14 | 28.19 | 18.07 | 27.15 | 34.23 | 22.77 | 28.05 |
| | LLM-Augmenter w/ Confidence | 28.16 | 18.74 | 22.26 | 28.46 | 18.88 | 27.42 | 33.24 | 22.55 | 27.31 |
| | **KALMV (Ours)** | **59.29** | **51.52** | **59.13** | **53.15** | **42.30** | **62.87** | **58.15** | **48.44** | **59.11** |

Table 2: **Results on WebQSP and Mintaka**, where we use Wikipedia as the knowledge source and report results with F1.

| Datasets | Methods | Base | Large | XL |
|---|---|---|---|---|
| **WebQSP** | Naive Language Models | 32.53 | 40.33 | 46.20 |
| | Knowledge-Augmented LMs | 27.96 | 27.39 | 26.40 |
| | Adaptive Retrieval w/ Confidence | 36.15 | 41.68 | 44.89 |
| | LLM-Augmenter w/ Knowledge F1 | 28.35 | 38.14 | 41.21 |
| | LLM-Augmenter w/ Confidence | 30.01 | 28.75 | 29.70 |
| | **KALMV (Ours)** | **56.70** | **60.63** | **63.75** |
| **Mintaka** | Naive Language Models | 16.16 | 20.90 | 26.99 |
| | Knowledge-Augmented LMs | 27.10 | 26.25 | 28.32 |
| | Adaptive Retrieval w/ Confidence | 24.74 | 26.20 | 28.87 |
| | LLM-Augmenter w/ Knowledge F1 | 29.84 | 40.30 | 43.87 |
| | LLM-Augmenter w/ Confidence | 28.81 | 27.64 | 30.91 |
| | **KALMV (Ours)** | **65.49** | **66.48** | **70.83** |

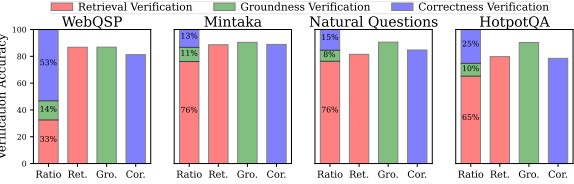

Figure 2: **Ratios of verification types and verification accuracies on them**, on each dataset with the FLAN Base as LMs.

of error-rectifying steps in the range of $\{1, 2, 3\}$, and filter out answers that are determined to have errors by our verifier after the maximum step. Further, for the ensemble, we use 5 different outputs, which have the probabilities of three choices (Section 3.2), from 5 different instructions, and average probabilities to select one option for verification.

# 5 Experimental Results and Analyses

**Main Results**   We conduct experiments on two question answering tasks: open-domain QA with Wikipedia and knowledge graph QA with Wikidata. As shown in Table 1, our proposed KALMV significantly improves the performance of knowledge-augmented LMs on all datasets across different LM sizes by effectively verifying errors in the knowl-

edge retrieval and answer generation steps. In addition, for knowledge graph QA, we also validate our KALMV on the setting where LMs are augmented with the documents from Wikipedia in Table 2, on which it also outperforms baselines substantially. Note that LLM-Augmenter, which verifies whether the generated answers are grounded in the retrieved knowledge, shows decent performance compared to other baselines. However, KALMV outperforms it by large margins, which suggests the importance of verifying the retrieval error and training the separate LM compared to using the heuristic measure to verify only the groundedness in answer generation.

**Analyses on Verification**   To understand how the proposed verifier works, we analyze it in multiple aspects. In the first bar of each subplot in Figure 2, we report the percentages of the knowledge retrieval error, the knowledge grounding error, and the correct generation, and we can see that the most common errors come from the incorrect knowledge

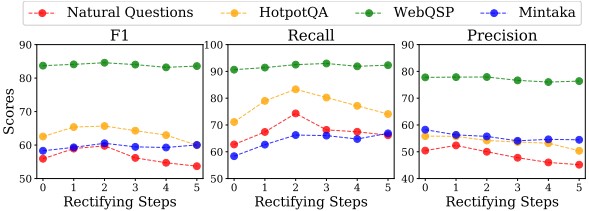

Figure 3: **Varying the number of rectifying steps**, on each dataset with F1, Recall, and Precision as the verifier metrics.

retrieval, which signifies the importance of verifying the retrieved knowledge. Also, on the remaining three bars in Figure 2, we report the verifier accuracy on each class category and then observe that our KALMV is able to detect errors in a balanced way across different verification categories.

We also report the performance of our verifier with regards to F1, recall, and precision scores in Figure 3, while varying the number of rectifying steps. In particular, precision denotes the proportion of the correct verification out of all verification predicted as correct; meanwhile, recall evaluates the proportion of the correctly predicted verification out of all actual correct verification. As shown in Figure 3, recall and F1 scores reach their almost highest points around two to three rectifying steps, while precision scores decrease slightly. These results suggest that, by increasing the number of rectifying steps, the coverage of our KALMV in delivering correct answers (i.e., recall) increases much, albeit with a slight compromise in the proportion of correct answers delivered (i.e., precision).

Please note that we also provide the **case study** on the three verification categories in Table 7.

**Ablation & Sensitive Analyses** To see how much our ensemble strategy contributes to the performance gain, and also how sensitive the components in KALMV are across different models, we perform ablation and sensitive analyses on ensemble, retrieval, verification, and generation parts. First, as shown in the first row of Table 3, ensemble, which forwards multiple verification instructions to the verifier and averages their results, improves the performance of both the verification and answer generation steps, demonstrating its efficacy.

For sensitive analyses, we first change the knowledge retriever for open-domain QA from the sparse (BM25) to the dense (DPR) retriever (Karpukhin et al., 2020). As shown in Table 3, while the dense retriever further brings performance improvement against the sparse retriever on most metrics, our KALMV consistently detects errors of knowledge-augmented LMs with high performance regardless

Table 3: **Ensemble and Sensitive Analyses on retrieval, verification, and generation**, on the Natural Questions data.

| Categories | Types | Verification | | Generation | |
|---|---|---|---|---|---|
| | | Acc | F1 | Acc | F1 |
| **Ensemble** | Yes | **78.39** | **55.91** | **50.43** | **52.98** |
| | No | 76.45 | 53.37 | 48.40 | 50.68 |
| **Retrieval Models** | BM25 | 78.39 | 55.91 | 50.43 | 52.98 |
| | DPR | 69.53 | **61.53** | **54.72** | **55.68** |
| **Verification LMs** | T5 (250M) | 76.23 | 50.00 | 42.33 | 44.63 |
| | FLAN (250M) | **78.39** | **55.91** | **50.43** | **52.98** |
| | ChatGPT | 65.71 | 43.17 | 33.16 | 36.68 |
| **Generation LMs** | T0 (3B) | 78.92 | 54.52 | 58.87 | 62.35 |
| | FLAN (3B) | **79.11** | **56.76** | 63.17 | 67.43 |
| | ChatGPT | 77.14 | 55.65 | **69.42** | **72.23** |

Table 4: **Results on Transfer Settings**, where our KALMV is trained on the Source dataset and tested on the Target dataset.

| Source | Target | F1 | EM | Acc |
|---|---|---|---|---|
| Natural Questions | Natural Questions | 52.98 | 42.36 | 50.43 |
| HotpotQA | Natural Questions | 56.26 | 46.70 | 53.02 |
| HotpotQA | HotpotQA | 64.06 | 52.31 | 55.84 |
| Natural Questions | HotpotQA | 55.08 | 42.17 | 45.56 |
| WebQSP | WebQSP | 74.31 | 63.92 | 77.78 |
| Mintaka | WebQSP | 69.86 | 60.00 | 72.47 |
| Mintaka | Mintaka | 59.29 | 51.52 | 59.13 |
| WebQSP | Mintaka | 48.06 | 40.25 | 46.19 |

of retrievers. Also, for sensitive analyses on verification and generation, we further include ChatGPT (OpenAI, 2022) as a reference model to understand the proprietary model's performance. Regarding verification, we observe that our FLAN-based instruction-finetuned verifier is superior to the ChatGPT (Peng et al., 2023), which suggests that customizing the available LM to our target verification task with further training is more worthwhile than using the general-purpose large LMs. Moreover, for generation LMs that make answers to the given questions, large LMs obviously outperform the performance of relatively small LMs, since large LMs might be more skilled and knowledgeable in answering questions. Note that our KALMV can accurately identify the errors even when coupled with ChatGPT as well as the other instruction-finetuned T0 (Sanh et al., 2022), confirming its versatility.

**Analyses on Generalization to Unseen Data** It is worthwhile noting that our KALMV can be directly applicable to other datasets without any further training on them. To show this, we first train the verifier of KALMV on the source data (e.g., Natural Questions) and then evaluate KALMV on the target data (e.g., HotpotQA), with FLAN Base used as the LM for generation and verification. As shown in Table 4, we observe that our KALMV has the capacity to generalize to other data without much performance degradation. Furthermore, for the Natural Questions dataset, the verifier trained

on the HotpotQA might be stronger than the verifier trained on the same Natural Questions, from the observation of the KALMV's performances on Natural Questions from models trained on each of HopotQA and Natrual Questions datasets, which further signifies its generalization ability.

# 6 Conclusion

In this work, we proposed Knowledge-Augmented Language Model Verification (KALMV), which identifies not only the relevance of the retrieved knowledge to the input query but also the faithfulness of the reflection of knowledge in the generated answers, in order to prevent incorrect answer generations with knowledge-augmented LMs. To this end, we developed a verifier that can detect errors in both the knowledge retrieval and answer generation stages by instruction-finetuning LMs. Further, during inference, we proposed to rectify errors by re-retrieving knowledge and re-generating answers if our KALMV detects errors, and also perform an ensemble over multiple verification outputs from different instructions, to improve the efficacy of the verifier. We validated KALMV on two question answering tasks and showed its effectiveness in significantly reducing hallucinations. We believe that KALMV will bring substantial practical impact in improving the reliability of LM-based systems, especially since it is a plug-and-play module.

## Limitations

In this section, we faithfully discuss the current limitations and potential avenues for future research.

First, we propose to instruction-finetune the verifier LM to customize it to the proposed verification task that aims to detect errors in knowledge retrieval and answer generation steps. Then, through our experimental results and analyses, we show that our proposed verifier trained by the automatically generated input-output pairs (See Section 3.2) is effective in identifying errors. However, the automatic label-generation processes that we suggest are indeed simple and they may introduce the potential to incorrectly generate the verification label in some particular scenarios (e.g., multi-step reasoning with multiple sources of knowledge). Therefore, someone may improve the labels required for instruction-finetuning verifiers by annotating them manually with humans or designing more sophisticated strategies, which we leave as future work.

Second, our work initiates a new problem setup

of detecting errors of knowledge-augmented LMs in two different perspectives: knowledge retrieval and answer generation. However, each component and strategy of the proposed KALMV method is a bit separated. Specifically, the retriever and verifier are not jointly trained, while the signal from training the verifier may help improve the retriever's performance. Also, regarding the error rectifying steps, while we can iteratively correct failures on knowledge-augmented LMs, the previous and current rectifying steps are handled separately. However, the current step may get benefits from the results of the previous steps. We leave developing and building more ideas on improving components of our proposed KALMV method as future work.

## Ethics Statement

Hallucination, which is a phenomenon where the language models generate responses that are plausible and sound yet factually incorrect, is a critical problem especially when deploying LMs in production since it can induce the spreading of misinformation. In this work, the proposed knowledge-augmented language model verification (KALMV) method contributes to significantly reducing hallucinations of LMs, by verifying their retrieved knowledge and generated answers, and further rectifying them if errors are detected. However, there may be some cases where our verifier misclassifies the failure cases of knowledge-augmented LMs as correct, potentially leading to severe negative consequences, especially in mission-critical domains and systems. Therefore, it is important for us to put more effort into making LMs more reliable and trustworthy with advanced verification methods.

## Acknowledgements

This work was supported by the Institute of Information & communications Technology Planning & Evaluation (IITP) grant funded by the Korea government (MSIT) (No. 2019-0-00075, Artificial Intelligence Graduate School Program (KAIST) and No. RS-2022-00187238, Development of Large Korean Language Model Technology for Efficient Pre-training), the National Research Foundation of Korea (NRF) grant funded by the Korea government (MSIT) (No. RS-2023-00256259), and the Engineering Research Center Program through the National Research Foundation of Korea (NRF) funded by the Korea Government (MSIT) (NRF-2018R1A5A1059921).

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

Table 5: **Relative Increment of Computational Costs**, which the verifier of our KALMV approach additionally yields, compared to the model (knowledge-augmented LMs with knowledge retrieval and answer generation) without the verification.

| Datasets | Base | Large | XL |
|---|---|---|---|
| WebQSP | 5.60% | 3.40% | 3.07% |
| Mintaka | 6.07% | 3.47% | 3.57% |
| Natural Questions | 10.51% | 9.26% | 6.54% |
| HotpotQA | 5.02% | 4.24% | 3.67% |

## A  Additional Experimental Setups

Here we provide additional experimental setups, including the instruction that we use for verification.

**Instruction Prompt**  In Table 6, we provide a set of 5 different instructions that we use for verification ensemble as well as instruction-finetuning verifiers (Please refer to Section 3.2 for details).

**LLM-Augmenter Details**  In our experiments in Section 5, we include this LLM-Augmenter model as our major baseline (Peng et al., 2023), and we now describe it in more detail. Note that the main focus of this baseline is to verify whether the generated answers from large LMs are grounded in the retrieved knowledge, and they propose two strategies to identify the groundedness. Specifically, the first strategy is the one that measures the Knowledge F1 score between the retrieved knowledge and the generated answer, which we already used for comparisons against our KALMV in our main experiments. On the other hand, the second strategy is to ask proprietary LMs (e.g., ChatGPT) to verify the groundedness of the generated answer in the retrieved knowledge. However, for the second one, it is infeasible for us to run every experiment with private Large LMs, and also it is clearly unfair to compare the public LMs against the proprietary LMs since their training data and capacity may be largely different. Nevertheless, we show that our KALMV is superior to LLM-Augmenter with ChatGPT on verification and answer generation in Table 3. Moreover, LLM-Augmenter with ChatGPT is known to have similar performances to the one that we compare (i.e, LLM-Augmenter w/ Knowledge F1) according to Peng et al. (2023), which may further support the fact that our KALMV is more effective on verification compared to the LLM-Augmenter based on ChatGPT since our KALMV significantly outperforms LLM-Augmenter w/ Knowledge F1.

## B  Additional Experimental Results

### B.1  Verification Cost

As it is worthwhile to investigate the increment of computational costs incurred by answer verification of our KALMV compared to the one without verification, we measure the relative increment in costs that our verifier additionally brings compared to the whole costs of running base knowledge-augmented LMs, and report it in Table 5. In particular, following the main experiment settings, we use the FLAN Base (250M) as the verification LM and use three different sizes of FLAN: Base (250M), Large (780M), and XL (3B), as the generation LM. Also, we set the cost of knowledge retrieval and answer generation (e.g., cost of running the entire knowledge-augmented LMs) as 100, and then report the relative increment from using the proposed verification. As shown in Table 5, our KALMV yields only the marginal increment, since not only do we use the smaller LM (Base) compared against larger LMs (Large and XL) for verification, but also the proposed verification LM generates only one token (e.g., A, B, or C) unlike the generation LM that decodes multiple tokens. For example, verifying answers with KALMV is 34 times faster than generating answers with Flan XL on the WebQSP data, which suggests that ours is highly efficient.

Yet, each rectifying step of our KALMV method incurs a cost that is approximately equivalent to the cost of running entire knowledge-augmented LMs with verification. To be specific, let's assume that, through the KALMV framework, the error in the generated answer is identified, the rectifying step is subsequently performed, and the new answer is verified as correct. Then, it takes twice as slow as the model without rectification. Yet, fortunately, since not every generated answer is verified as incorrect, the number of samples that require rectifying steps is far less than the number of all samples (e.g., only 38% of samples require rectification on WebQSP).

### B.2  Case Study

In Table 7, we provide examples of our KALMV framework on three verification categories: incorrect knowledge retrieval, incorrect answer generation, and correct answer generation, on knowledge-augmented LMs. As shown in Table 7, KALMV can detect the errors of knowledge-augmented LMs by contextualizing and understanding the relationships between the input question, retrieved knowledge, and generated answer effectively.

Table 6: A list of instructions that we use for verification with ensemble as well as for instruction-finetuning verifiers. Note that the variable inside the set parentheses {} is replaced with its actual string (e.g., input question, knowledge, and generated output).

| Indices | Instructions |
|---|---|
| 1 | The following is a multiple choice question about a question answering task. In this task, you should generate an output given a question with a passage. The passage is retrieved from Wikipedia, which may or may not be helpful to answer the question. 
 Question: {question} 
 Passage: {passage} 
 Output: {answer} 
 Options: 
 A. The passage is unhelpful to answer the question. 
 B. The passage is helpful to answer the question, yet the generated output for the question is incorrect. 
 C. The generated output for the question is correct. 
 Select one option: |
| 2 | Question: {question} 
 Passage: {passage} 
 Output: {answer} 
 Options: 
 A. The passage is unhelpful to answer the question. 
 B. The passage is helpful to answer the question, yet the generated output for the question is incorrect. 
 C. The generated output for the question is correct. 
 Select one option: |
| 3 | Given a question and a passage from Wikipedia, you should generate an output as follows: 
 Question: {question} 
 Passage: {passage} 
 Output: {answer} 
 This is a multiple choice question, and, based on the above information, you need to select one option among three, as follows: 
 A. The passage is unhelpful to answer the question. 
 B. The passage is helpful to answer the question, yet the generated output for the question is incorrect. 
 C. The generated output for the question is correct. 
 Select one option: |
| 4 | Here is a question, passage, and generated output from the question and passage. Based on them, you need to select one option among the three. 
 Question: {question} 
 Passage: {passage} 
 Output: {answer} 
 Options: 
 A. The passage is unhelpful to answer the question. 
 B. The passage is helpful to answer the question, yet the generated output for the question is incorrect. 
 C. The generated output for the question is correct. 
 Select one option: |
| 5 | Given a question, passage, and output, which option is the best? 
 Question: {question} 
 Passage: {passage} 
 Output: {answer} 
 Options: 
 A. The passage is unhelpful to answer the question. 
 B. The passage is helpful to answer the question, yet the generated output for the question is incorrect. 
 C. The generated output for the question is correct. 
 Select one option: |

Table 7: Examples of three types of verification outputs, such as retrieval error, generation error, and correct answer of knowledge-augmented LMs, determined by our KALMV on the Natural Question dataset with FLAN Base as the generation LM.

| Types | Examples |
|-------|----------|
| Retrieval Error | **Question**: who sang the song good morning good morning?
**Knowledge**: Good Morning Call
**Correct answers**: ['Gene Kelly', "Donald O'Connor", 'Judy Garland', 'Debbie Reynolds', 'Mickey Rooney']
**Generated answer**: The Beatles |
| Retrieval Error | **Question**: when did taylor swift's first album release?
**Knowledge**: 1989 is the fifth studio album by American singer-songwriter Taylor Swift released on October 27, 2014, through Big Machine Records. Swift began composing the album following release of previous studio effort, Red (2012). Over the course of the two-year songwriting period, she collaborated with producers Max Martin and Shellbackâ€"Martin served as the album's executive producer alongside Swift. The album's title was named after the singer's birth year and inspired by the pop music of the 1980s.
**Correct answers**: ['October 24, 2006', '2005']
**Generated answer**: October 27, 2014 |
| Retrieval Error | **Question**: who sang i ran all the way home?
**Knowledge**: In 2007, the song was covered by Paul McCartney who sung it, and Allen Toussaint playing the piano, as their contribution to Goin' Home: A Tribute to Fats Domino (Vanguard).
**Correct answers**: ['The Impalas']
**Generated answer**: Paul McCartney |
| Generation Error | **Question**: what is the hot coffee mod in san andreas?
**Knowledge**: Hot Coffee is a normally inaccessible mini-game in the 2004 video game Grand Theft Auto: San Andreas, developed by Rockstar North. Public awareness of the existence of the mini-game arrived with the release of the Hot Coffee mod, created for the Microsoft Windows port of GTA: San Andreas in 2005. This mod enables access to the mini-game.
**Correct answers**: ['a normally inaccessible mini-game']
**Generated answer**: enables access to the mini-game |
| Generation Error | **Question**: where does a roadrunner live in the desert?
**Knowledge**: Roadrunners inhabit the deserts of the southwestern United States, Mexico, and Central America. They live in arid lowland or mountainous shrubland, widely dispersed in dry open country with scattered brush. They are non-migratory, staying in their breeding area year-round.[10] The greater roadrunner is not currently considered threatened in the US, but is habitat-limited.
**Correct answers**: ['mountainous shrubland', 'arid lowland']
**Generated answer**: United States |
| Generation Error | **Question**: what is the angle of the tower of pisa?
**Knowledge**: The height of the tower is 55.86 metres (183.27 feet) from the ground on the low side and 56.67 metres (185.93 feet) on the high side. The width of the walls at the base is 2.44 m (8 ft 0.06 in). Its weight is estimated at 14,500 metric tons (16,000 short tons). The tower has 296 or 294 steps; the seventh floor has two fewer steps on the north-facing staircase. Prior to restoration work performed between 1990 and 2001, the tower leaned at an angle of 5.5 degrees, but the tower now leans at about 3.99 degrees. This means the top of the tower is displaced horizontally 3.9 metres (12 ft 10 in) from the centre.
**Correct answers**: ['about 3.99 degrees', '3.99 degrees', 'at about 3.99 degrees']
**Generated answer**: 5.5 degrees |
| Correct Answer | **Question**: who wrote he ain't heavy he's my brother lyrics?
**Knowledge**: "He Ain't Heavy, He's My Brother" is a popular music ballad written by Bobby Scott and Bob Russell. Originally recorded by Kelly Gordon in 1969, the song became a worldwide hit for The Hollies later that year and again for Neil Diamond in 1970. It has been covered by many artists in subsequent years. The Hollies' and Rufus Wainwright's versions of the song were featured in the film Zoolander.
**Correct answers**: ['Bobby Scott', 'Bob Russell']
**Generated answer**: Bobby Scott and Bob Russell |
| Correct Answer | **Question**: who is the lead singer of depeche mode?
**Knowledge**: David Gahan (born David Callcott; 9 May 1962) is an English singer-songwriter, best known as the baritone lead singer of the electronic band Depeche Mode since their debut in 1980. He is also a successful solo artist, releasing albums in 2003 (Paper Monsters) and 2007 (Hourglass).
**Correct answers**: ['David Gahan']
**Generated answer**: David Gahan |

| Types | Examples |
|---|---|
| Correct Answer | **Question**: when was the first hunger games book published? |
| | **Knowledge**: The Hunger Games was first published in hardcover on September 14, 2008, by Scholastic, featuring a cover designed by Tim O'Brien. It has since been released in paperback and also as an audiobook and ebook. After an initial print of 200,000, the book had sold 800,000 copies by February 2010. Since its release, The Hunger Games has been translated into 26 languages, and publishing rights have been sold in 38Â territories. The novel is the first in The Hunger Games trilogy, followed by Catching Fire (2009) and Mockingjay (2010). A film adaptation, directed by Gary Ross and co-written and co-produced by Collins herself, was released in 2012. |
| | **Correct answers**: ['September 14, 2008', '2008'] |
| | **Generated answer**: September 14, 2008 |