# OpenReview forum: "Knowledge-Augmented Language Model Verification"
_EMNLP/2023/Conference — EMNLP 2023 Main_

### Official Review · Reviewer_rWYL · 2023-08-03

**Soundness:** 4

**Excitement:**

3: Ambivalent: It has merits (e.g., it reports state-of-the-art results, the idea is nice), but there are key weaknesses (e.g., it describes incremental work), and it can significantly benefit from another round of revision. However, I won't object to accepting it if my co-reviewers champion it.

**Paper Topic And Main Contributions:**

The authors propose a system for validating the 1) relevance and 2) faithfulness of a language models response with respect to retrieved knowledge. Their approach solves questions by first retrieving knowledge from a knowledge base, formulating a response based on that retrieved knowledge, and then validating the response using an instruction-finetuned verifier language model that can identify whether a response fits 1 and 2 (above). They evaluate their approach on a number of datasets and achieve better performance as compared to other retrieval-based approaches

**Questions For The Authors:**

It would be helpful to include the state-of-the-art for each dataset (not just limited to the baselines here). While it is understandable to group and compare against similar approaches, it is hard for me to tell how well the model performs when comparing against such a restricted set  of methods

**Reasons To Accept:**

The emphasis on validating the faithfulness of a model's "explanation" (i.e., retrieved knowledge here, but it could be other outputs like chain-of-thought) is an important topic

The paper is well structured and easy to follow

The results as compared to similar approaches seem reasonable and a good number of datasets were explored.

**Reasons To Reject:**

The novelty of the approach (as I'm understanding it) appears to be the use of a fine tuned language model to verify a response against retrieved facts. This is a very straightforward approach that may have been unexplored for this particular task (i.e., question-answering against a KB) but not overall (the use of LLMs as verifiers is not new). To me, this makes the main contribution of this work very incremental

It seems to me that the need to fine-tune the verifier would be quite limiting, as you could then only apply this to datasets for which you had adequate amounts of data

**Reproducibility:**

4: Could mostly reproduce the results, but there may be some variation because of sample variance or minor variations in their interpretation of the protocol or method.

**Reviewer Confidence:**

3: Pretty sure, but there's a chance I missed something. Although I have a good feel for this area in general, I did not carefully check the paper's details, e.g., the math, experimental design, or novelty.

---

> ### Author Rebuttal · Authors · 2023-08-28
>
> We sincerely appreciate your constructive and helpful comments. During the response period, we have made every effort to faithfully address all your comments below.
>
> ---
>
> > **Weakness 1:** The novelty of the approach appears to be the use of a fine-tuned LM to verify a response against retrieved facts, which is a very straightforward approach. It may have been unexplored for this particular task (i.e., question-answering against a KB) but not overall (i.e., the use of LLMs as verifiers is not new), which makes this work incremental.
>
> Our contribution is not limited to the mere fine-tuning of LMs for response verification, and the proposed output verifications of **knowledge-augmented LMs by considering the knowledge retrieval and grounding errors** are substantially different from existing approaches that verify the outputs of **LMs without considering knowledge augmentation**.
>
> Specifically, as described in Lines 55 - 68 and Lines 207 - 235, existing verifiers, which include the usage of the external knowledge for verification, suffer from the following two problems: 1) the model may fail to retrieve the relevant knowledge to the query and 2) the generated answer may not be grounded in the retrieved knowledge. However, existing approaches overlook these challenges and rather suppose that there are no retrieval and grounding errors during answer generation. In other words, they focus on validating the generated answer itself by referring to the external knowledge, instead of considering more fundamental problems of the knowledge retrieval and grounding errors.
>
> In contrast to such existing works, we point out the aforementioned two prevalent yet underexplored challenges of knowledge-augmented LMs (i.e., retrieval and grounding errors) and then tackle them by instruction-finetuning the verifier along with the proposed ensemble and answer rectifying strategies. More technically speaking, existing verification methods use **a pair of the input query and the generated answer** to verify its validity and do not check the retrieval and generation errors even if augmented with external knowledge. Meanwhile, our verification method uses **a triplet of input query, retrieved knowledge, and generated answer** to verify both the retrieval and generation errors with respect to the knowledge.
>
> Moreover, the existing verification method that uses only LLMs without external knowledge is limited in its capacity, as demonstrated in Table 3, since the knowledge internalized in parameters may be incomplete, inaccurate, and outdated (i.e., LLMs cannot store all the world knowledge).
>
> Therefore, we strongly believe that providing such new perspectives on handling retrieval and grounding errors for verifying the knowledge-augmented LMs unlike existing verifiers, and then tackling them with the new training strategies is not an incremental but rather a substantial contribution.
>
> ---
>
> > **Weakness 2:** It seems to me that the need to fine-tune the verifier would be quite limiting, as you could then only apply this to datasets for which you had adequate amounts of data.
>
> **Our instruction-finetuned verifier that is trained on a certain dataset is directly generalizable to other datasets even without training on them**, i.e., our verifier is not limited to datasets that are used for verifier training. Specifically, we additionally conduct experiments on transfer learning, where we first train the verifier on the source domain (e.g., Natural Questions) and then generalize it on the target domain (e.g., HotpotQA) without any training samples on the target dataset, with FLAN Base as the generation and verification LMs.
>
> As shown in Table C.1 below, we observe **that the verifier trained on the source dataset can be directly used for the target dataset without significant performance degradations**. Also, for the Natural Questions dataset, we observe that the verifier trained on the HotpotQA dataset is superior to the verifier trained on the Natural Questions. These results demonstrate that our instruction-finetuned verifier is not limited to the dataset having an adequate amount of training data, and rather is broadly generalizable across different datasets even without training samples on them. We will include the results in the revision.
>
> Table C.1: Results where the verifier is trained on Source data and tested on Target data.
> |Source|Target|F1|EM|Acc|
> |---|---|---|---|---|
> |Natural Questions|Natural Questions|52.98|42.36|50.43|
> |HotpotQA|Natural Questions|56.26|46.70|53.02|
> ||
> |HotpotQA|HotpotQA|64.06|52.31|55.84|
> |Natural Questions|HotpotQA|55.08|42.17|45.56|
> ||
> |WebQSP|WebQSP|74.31|63.92|77.78|
> |Mintaka|WebQSP|69.86|60.00|72.47|
> ||
> |Mintaka|Mintaka|59.29|51.52|59.13|
> |WebQSP|Mintaka|48.06|40.25|46.19|
>
> Additionally, we would like to point out that **using general-purpose LLMs that are not tailored to verifying the knowledge-augmented LMs is inferior to our instruction-finetuned smaller LM**, as shown in Table 3. This result further demonstrates that our technique for fine-tuning the smaller verifier is not limiting but rather has huge advantages in both performance and efficiency.
>
> ---
>
> > **Question 1:** It would be helpful to include the state-of-the-art for each dataset. While it is understandable to group and compare against similar approaches, it is hard for me to tell how well the model performs when compared against such a restricted set of methods.
>
> Thank you for your suggestion. However, we would like to emphasize that including the state-of-the-art knowledge-augmented LMs in our experiments is unnecessary, since **we already use the architectures of one of the most powerful and very recently proposed knowledge-augmented LMs** (Mallen et al., 2023; Back et al., 2023). Also, **our verification method can orthogonally improve any knowledge-augmented LMs** including the most powerful one.
>
> Specifically, existing knowledge-augmented LMs aim at improving the **architectures of knowledge retrieval and answer generation methods**; meanwhile, our work aims at **detecting the retrieval and generation errors of any existing retrievers and generators**, including the most powerful ones, in order to improve their reliability rather than directly improving their effectiveness with model changes.
>
> Additionally, including and comparing against models that use different LM backbones (e.g., FLAN vs GPT-4) may not provide any valuable experimental insights, since different LMs have different capacities which makes apple-to-apple comparisons between them infeasible.

---

### Official Review · Reviewer_QSUP · 2023-08-04

**Soundness:** 4

**Excitement:**

3: Ambivalent: It has merits (e.g., it reports state-of-the-art results, the idea is nice), but there are key weaknesses (e.g., it describes incremental work), and it can significantly benefit from another round of revision. However, I won't object to accepting it if my co-reviewers champion it.

**Paper Topic And Main Contributions:**

This paper proposes a method to verify and correct the retrieval error and generation error of language models that are augmented with external knowledge. The paper also uses an ensemble of different instructions and iteratively corrects the errors until the verifier confirms the correctness of the retrieved knowledge and the generated answer.

**Questions For The Authors:**

- Is there any kind of guarantee the answers having correct integration of retrieved results present in the Top k answer without touching upon the core mechanism of model generation? A more convincing approach is to further ensure the groundness of the next generation after a generating error show up.
- The paper claims that the input-output pair is automatically constructed, how the label of groundedness of the augmented answer was assigned automatically? This is the point that needs to be clarify.

**Reasons To Accept:**

The paper is well-written and organized and provides a clear motivation and problem formulation. This work has a simple idea but good results.

**Reasons To Reject:**

- Is there any kind of guarantee the answers having correct integration of retrieved results present in the Top k answer without touching upon the core mechanism of model generation? A more convincing approach is to further ensure the groundness of the next generation after a generating error show up.
- The paper claims that the input-output pair is automatically constructed, how the label of groundedness of the augmented answer was assigned automatically? This is the point that needs to be clarify.
- And I did not find the description about whether these baselines use the same generated model, which is important of comparison in the experiment.
- The proposed method is more like expand the space to choose the answer by the help of a trained verifier. So is there a more integreted and concise structure to build this model. For example, aligning directly during retrieval and generation to ensure the correctness of each step.

**Reproducibility:**

4: Could mostly reproduce the results, but there may be some variation because of sample variance or minor variations in their interpretation of the protocol or method.

**Reviewer Confidence:**

4: Quite sure. I tried to check the important points carefully. It's unlikely, though conceivable, that I missed something that should affect my ratings.

---

> ### Author Rebuttal · Authors · 2023-08-28
>
> We sincerely appreciate your constructive and helpful comments. During the response period, we have made every effort to faithfully address all your comments below.
>
> ---
>
> > **Weakness 1.1:** Is there any kind of guarantee that the answers, having correct integration of retrieved results, present in the Top-K answers without touching upon the core mechanism of model generation?
>
> While it is not always guaranteed that one of the Top-K answers includes the correct knowledge retrieved from the KB, iteratively generating the answers until making the correct one by using Top-K sampling is still highly effective and efficient. Specifically, ensuring that the answer is grounded in the correct knowledge is not straightforward and rather a substantially difficult task, since it not only requires the verifier to check whether the generator uses one of the retrieved documents/facts but also to identify whether the generator selects only the correct portion of the retrieved document among all the other information. Therefore, we simply sample one candidate answer among the Top-K answers at every rectifying step, and, by increasing the number of K (i.e., increasing the answer space), the model may generate the answer with a sufficient number of rectifying steps.
>
> Also, while touching upon the core mechanism of model generation to further ensure the grounding of the generated answer in the retrieved knowledge is not the scope of our work (i.e., our target is on building a post hoc verifier rather than improving the generator) and it might be a promising avenue for future work, it has a critical drawback. In particular, one advantage of our verification method is that we only use the input and output pair (e.g., question, retrieved knowledge, and generated answer) of the answer generator, which allows us to use even the proprietary LLMs that are accessible via API calls. However, the method, which modifies the answer generation strategy, is not applicable to such private LLMs.
>
> ---
>
> > **Weakness 1.2:** A more convincing approach is to further ensure the groundness of the next generation after a generation error shows up.
>
> Please see the response to Weakness 1.1 for detailed answers. In summary, ensuring the groundness of the generated text is an extremely difficult problem since the generator further recognizes and selects the right portion of the retrieved knowledge, which is much more complex than mere copy-and-paste of the retrieved knowledge. Also, for private LMs that are accessible via API calls, we may not explicitly ensure the groundness during the answer generation; meanwhile, our verification approach can ensure it in a post hoc manner.
>
> ---
>
> > **Weakness 2:** It is unclear how the label of groundedness of the generated answer was assigned automatically based on the input-output pairs.
>
> As explained in Lines 374 - 377, **when the generated answer does not have the overlapping tokens with the retrieved correct knowledge**, we label it as a grounding (generation) error. This is because, in such cases, the generation LM does not utilize the retrieved knowledge even if the retrieved knowledge contains the correct answer. Please note that, as discussed in footnote 1 as well as Lines 662 - 678 of the Limitations section, while our automatic label generation strategy works well (Section 5) despite its simplicity, there may exist more sophisticated ways to automatically generate the labels, which we leave as future work.
>
> ---
>
> > **Weakness 3:** I did not find the description about whether these baselines use the same generated model, which is important for comparison in the experiment.
>
> We apologize for the confusion. As described in Section 4.4 (Lines 520 - 531), for all the knowledge-augmented LM baselines and our method, we use the same retriever, the same generator, and the same prompt for augmenting LMs, which we will clarify in the revision.
>
> ---
>
> > **Weakness 4.1:** The proposed method is more like expanding the space to choose the answer with the verifier.
>
> Yes, one contribution of our work is to continuously expand the answer space by retrieving the relevant knowledge and generating the answer until the knowledge-augmented LM reaches the correct answer, based on our verification and error rectification strategies.
>
> ---
>
> > **Weakness 4.2:** There may exist a more integrated and concise structure to build the verification method that can expand the search space, e.g., aligning directly during retrieval and generation to ensure the correctness of each step.
>
> Yes, we also believe that there may exist a more improved way of building the verification method. However, our work paves a new avenue for verifying and rectifying two types of errors on knowledge-augmented LMs, which have been underexplored so far; thus, further improving them with additional verification strategies would be promising future work. Also, please note that the suggested direction of aligning the retrieval with the generation may not be applicable to LLMs that are accessible via API calls, since we cannot manipulate the generation process of such LLMs and the retriever may not be jointly trainable with them.
>
> ---
>
> > **Question 1 & Question 2:**
>
> Please refer to our responses for Weakness 1 and Weakness 2 above for our answers to these questions.

---

### Official Review · Reviewer_Fsm3 · 2023-08-05

**Soundness:** 4

**Excitement:**

4: Strong: This paper deepens the understanding of some phenomenon or lowers the barriers to an existing research direction.

**Paper Topic And Main Contributions:**

This paper introduces a small, tailorable LM verifier to improve factuality of knowledge-augment LMs, particularly in checking whether the LM: 1. retrieves the relevant knowledge, 2. incorporates the retrieved knowledge appropriately in its generations. The verifier is trained via instruction fine-tuning to perform both types of verification. Evaluated on 4 datasets (Natural Questions, HotpotQA, WebQSP, Mintaka), the method in the paper (KALMV) significantly outperforms retrieval-based LMs without verification.

**Questions For The Authors:**

1. I’m not quite sure what it means to evaluate the verification accuracy of generation LMs and generation accuracy of verification LMs (Table 3). Can you please clarify how this works and what the takeaway from these numbers is supposed to be?
2. “Therefore, we design various instructions, forward them to our single verifier, and ensemble the multiple outputs from the verifier with average” (L392-394). Are you instruction-fine-tuning with each of these five different instructions?


**Reasons To Accept:**

1. The paper is well-written.
2. The method presented in the paper is simple to understand and empirically effective, demonstrating significant improvement across 4 domains compared to multiple retrieval-based baselines without verification.
3. Detailed analysis allows drawing further insight into the source of the gains: e.g. most retrieval errors come from retrieving irrelevant facts/documents, ensembling results of different verifier prompts is useful, etc.


**Reasons To Reject:**

1. The analysis could be improved with a plot of rectifying steps vs. end-answer accuracy / F1 / precision / recall. Currently I’m not quite sure what the takeaway is supposed to be from the current plots (Fig. 3) of rectifying steps vs. verifier performance. It seems that verifier performance as a function of rectifying steps is likely more a function of base retrieval/answer accuracy after those steps, rather than something fundamentally changing with the verifier itself. Does end-answer recall peak after 2 rectifying steps? Why does it drop after 2 rectifying steps?
2. I’m not sure if including adaptive retrieval (L479 / Table 1) makes sense as a strong baseline here. It seems likely to be strictly worse than knowledge-augmented LMs as it is not retrieving for every query (indeed this seems supported by empirical results in Table 1) – its usage seems to be predominantly in improving efficiency of retrieval LMs by allowing the LM not to have to retrieve for every query.
3. The one drawback I can imagine with applying this technique in practice is the relative expense of having to verify and rectify errors at each step, following both retrieval and answer generation. This is not necessarily an indictment of this technique in general, as there are arguably settings where factuality matters above all, regardless of expense. However, perhaps the computational expense of this technique can be quantified for future reference.


**Reproducibility:**

4: Could mostly reproduce the results, but there may be some variation because of sample variance or minor variations in their interpretation of the protocol or method.

**Reviewer Confidence:**

4: Quite sure. I tried to check the important points carefully. It's unlikely, though conceivable, that I missed something that should affect my ratings.

**Typos Grammar Style And Presentation Improvements:**

L584-590: Figure 2 → Figure 3

---

> ### Author Rebuttal · Authors · 2023-08-29
>
> We sincerely appreciate your constructive and helpful comments. During the response period, we have made every effort to faithfully address all your comments below.
>
> ---
>
> > **Weakness 1.1:** The analysis in Figure 3 (i.e., varying the number of rectifying steps for verification) could be improved, by reporting the performance (e.g., accuracy, F1, precision) of generated answers instead of reporting the verification performance.
>
> We would like to clarify that, in Figure 3, **precision already captures the answer generation performance** (i.e., precision denotes the proportion of correctly generated outputs out of all generated outputs that are predicted as correct). On the other hand, **recall captures the answer verification performance** (i.e., recall denotes the proportion of correctly verified and correctly generated outputs out of all outputs that are labeled as correct). Therefore, Figure 3 shows both the answer generation and the answer verification performances.
>
> ---
>
> > **Weakness 1.2:** I’m not quite sure about the takeaway from Figure 3.
>
> We apologize for the confusion. In order to understand the key takeaway of Figure 3, we need to jointly look at the results on both precision and recall. As explained in the response to Weakness 1.1, precision captures the answer generation accuracy; meanwhile, recall captures the coverage of the knowledge-augmented LMs in delivering the correct outputs. Therefore, if the precision is low and the recall is high, the number of times that the user receives the correct outputs (recall) is increased, while the proportion of correct outputs on all the received outputs (precision) is decreased, i.e., the coverage is high yet the reliability is low.
>
> Regarding the takeaway from Figure 3, when increasing the number of rectifying steps, the recall tends to increase while the precision tends to decrease, which suggests that our KALMV can provide more numbers of correct responses to the user while the user may experience reduced reliability. Thus, **Figure 3 shows the trade-off between the coverage and the reliability with respect to the number of rectifying steps**, and perhaps can further suggest that using 1 or 2 rectifying steps may offer the optimal balance between them.
>
> ---
>
> > **Weakness 1.3:** It seems that, in Figure 3, reporting the verifier performance as the function of rectifying steps does not show something that is fundamentally changing.
>
> Please see our responses to Weakness 1.1 and Weakness 1.2. The performance of the verifier on precision and recall with respect to the number of rectifying steps shows the fundamental changes in model behaviors (e.g., the trade-off between the reliability and the coverage), beyond the point that it can rectify the retrieval and generation errors.
>
> ---
>
> > **Weakness 1.4:** In Figure 3, does the end-answer recall peak at 2 rectifying steps? Why does it drop after 2?
>
> As shown in Figure 3, the end-answer recall drops after 2 rectifying steps on most datasets. This is because a large portion of errors comes from retrieval (Figure 2) and the retrieved knowledge becomes less relevant to the input query after several rectification steps (e.g., we should retrieve and use the top-5 ranked knowledge if retrieval fails 4 times), making more incorrect answers. Consequently, when the number of rectifying steps is large, since more numbers of incorrect answers may be filtered out by our verifier due to their noise and fundamental errors, the recall (coverage) may decrease after 2 rectifying steps.
>
> ---
>
> > **Weakness 2:** I’m not sure whether including the adaptive retrieval (Mallen et al., 2023) makes sense as a strong baseline since it does not perform retrieval for every query, which seems likely to be worse than knowledge-augmented LMs that always perform retrieval.
>
> We include this adaptive retrieval method (Mallen et al., 2023) as a baseline, since it indeed contributes to improving the performance of knowledge-augmented LMs by adaptively retrieving the knowledge. In other words, this work shares a similar focus to our work that aims at improving the knowledge-augmented LMs while exploring a different approach - adaptive retrieval, unlike our approach to verification. Therefore, for this shared reason for improving the knowledge-augmented LMs, we consider adaptive retrieval as a fair and valid baseline.
>
> Specifically, as shown in Figure 1 of adaptive retrieval (Mallen et al., 2023), it outperforms the naive knowledge-augmented LMs when the sizes of LMs are relatively large, which confirms that it is a strong baseline. Also, in Table 1 of our work, we also observe that the adaptive retrieval baseline consistently outperforms the naive knowledge-augmented LMs on Flan XL (3B) in most settings; again, the adaptive retrieval models serve as clear baselines. However, based on the comparison between this adaptive retrieval and our verification method in Table 1, both of which aim at enhancing knowledge-augmented LMs, we demonstrate that our verification strategy is superior to adaptive retrieval in performance, which further signifies the importance of our verification for the knowledge-augmented LMs.
>
> ---
>
> > **Weakness 3.1:** One drawback with applying the proposed KALMV in practice is the relative expense of having to verify and rectify errors at each step along with both retrieval and answer generation, which is not necessarily an indictment though since there are arguably settings where factuality matters.
>
> As you pointed out, we strongly believe that, in many practical applications (e.g., mission-critical domains such as biomedicine), **providing factually correct answers in response to user’s queries might be much more beneficial at the expense of using some computational costs** than providing unreliable answers promptly without verification. Also, please note that, while we use extra costs for verifying answers and rectifying errors (See response to Weakness 3.2 below), the proposed KALMV incomparably outperforms existing knowledge-augmented LMs without verification, as well as we use the smaller LM (250M) for verification (i.e., the verification cost is indeed marginal compared to answer generation).
>
> ---
>
> > **Weakness 3.2:** The computational expense of the proposed KALMV on answer generation and verification can be further quantified, for future reference.
>
> Thank you for your suggestion. We additionally measure the relative increment in the computational costs for answer verification, compared to the model without verification.
>
> In particular, following the main experiment settings in our work, we use the FLAN base (250M) as the verification LM and use three different variants: Base (250M), Large (780M), and XL (3B), as the generation LM. Also, we set the cost of knowledge retrieval and answer generation (e.g., cost of running the entire knowledge-augmented LMs) as 100, and then report the relative computational cost increment from using the proposed verification. As shown in Table A.1, our KALMV brings only the marginal computational cost increment, since not only do we use the smaller LM (Base) compared against larger LMs (Large and XL) for verification, but also the proposed verification LM generates only one token (See Lines 352 - 359) unlike the generation LM that decodes multiple tokens. For example, our verification model is 34 times faster than generating the answer with Flan XL on the WebQSP dataset, which suggests that our proposed verification method is highly efficient.
>
> Table A.1: Relative computational cost on verification against knowledge retrieval and answer generation of knowledge-augmented LMs, where we use the Flan Base for verification.
> |Data|Base|Large|XL|
> |---|---|---|---|
> |WebQSP|5.60%|3.40%|3.07%|
> |Mintaka|6.07%|3.47%|3.57%|
> |Natural Questions|10.51%|9.26%|6.54%|
> |HotpotQA|5.02%|4.24%|3.67%|
>
> On the other hand, each rectifying step incurs a cost approximately equivalent to that of performing initial knowledge-augmented LMs with verification. For instance, if the error is identified, the rectifying step is performed, and the subsequent answer is verified as correct, it takes twice as long. Fortunately, since not every generated answer is verified as incorrect, the number of samples that require rectifying steps is far less than the number of all samples (e.g., only 38% of samples require rectification on WebQSP). We will include the results and discussions on cost in the revision.
>
> ---
>
>
> > **Question 1.1:** I’m not quite sure what it means to evaluate the verification accuracy of generation LMs and generation accuracy of verification LMs (Table 3). Can you clarify it?
>
> We apologize for the confusion. The leftmost column in Table 3 represents the specific category for analyses. For example, the verification accuracy of the “Generation LMs” category denotes the accuracy of the verification LM where the generation LM is selected among T0 (3B), FLAN (3B), and ChatGPT. Similarly, the generation accuracy of the “Verification LMs” category denotes the accuracy of the generation LM where the verification LM varies across T5 (250M), FLAN (250M), and ChatGPT.
>
> ---
>
> > **Question 1.2:** What is the takeaway from the results in Table 3?
>
> The main purpose of experiments in Table 3 is to vary the models on each category, in order to perform either ablation studies or sensitivity analyses on each component of our KALMV. For example, in the “Verification LMs” category, we validate the verification performances of three different models, such as T5 (250M), FLAN (250M), and ChatGPT, and then show that the instruction-finetuned tailorable LM (FLAN) is superior to ChatGPT on verification.
>
>
> ---
>
> > **Question 2:** Do you instruction-finetuning the model with each of these five different instructions (Lines 392 - 394)?
>
> Yes, your understanding is correct. Specifically, during training, we instruction-finetune the single verifier with each of the five verification instructions. Then, during inference, we forward those five instructions to the trained verifier and ensemble their results to produce the output.
>
> ---
>
> > **Typo:** In Lines 584-590, Figure 2 → Figure 3.
>
> Thank you for correcting it.

---

### Meta-Review · Area_Chair_u8jh · 2023-09-16

**Recommendation:** 5

**Metareview:**

The authors propose a verifier for RAG models that verifies that the retrieved evidence is relevant to the query and that the generation is attributed in the evidence.

The reviewers agree that the tackled problem is important, and that the proposed approach is simple and effective.

The reviewers also note that using an LLM for verification for this specific task is relatively incremental, as using verifiers in NLP is well known. In addition, the paper should be edited for further clarification regarding specific questions raised by the reviewers.

---

### Decision · Program_Chairs · 2023-10-07

**Decision:**

Accept-Main

**Comment:**

The authors propose a verifier for RAG models that verifies that the retrieved evidence is relevant to the query and that the generation is attributed in the evidence.

The reviewers agree that the tackled problem is important, and that the proposed approach is simple and effective.

The reviewers also note that using an LLM for verification for this specific task is relatively incremental, as using verifiers in NLP is well known. In addition, the paper should be edited for further clarification regarding specific questions raised by the reviewers.